# A Machine-Learning-Based Analysis of the Relationships between Loneliness Metrics and Mobility Patterns for Elderly

**DOI:** 10.3390/s22134946

**Published:** 2022-06-30

**Authors:** Aditi Site, Saigopal Vasudevan, Samuel Olaiya Afolaranmi, Jose L. Martinez Lastra, Jari Nurmi, Elena Simona Lohan

**Affiliations:** 1Faculty of Information Technology and Communication Sciences, Tampere University, 33720 Tampere, Finland; jari.nurmi@tuni.fi (J.N.); elena-simona.lohan@tuni.fi (E.S.L.); 2Faculty of Engineering and Natural Sciences, Tampere University, 33720 Tampere, Finland; saigopal.vasudevan@tuni.fi (S.V.); samuel.afolaranmi@tuni.fi (S.O.A.); jose.martinezlastra@tuni.fi (J.L.M.L.)

**Keywords:** indoor mobility, outdoor mobility, machine learning, loneliness, XGBoost, random forest, support vector machines, classification, senior citizens, UCLA score, Lubben score

## Abstract

Loneliness and social isolation are subjective measures associated with the feeling of discomfort and distress. Various factors associated with the feeling of loneliness or social isolation are: the built environment, long-term illnesses, the presence of disabilities or health problems, etc. One of the most important aspect which could impact feelings of loneliness is mobility. In this paper, we present a machine-learning based approach to classify the user loneliness levels using their indoor and outdoor mobility patterns. User mobility data has been collected based on indoor and outdoor sensors carried on by volunteers frequenting an elderly nursing house in Tampere region, Finland. The data was collected using Pozyx sensor for indoor data and Pico minifinder sensor for outdoor data. Mobility patterns such as the distance traveled indoors and outdoors, indoor and outdoor estimated speed, and frequently visited clusters were the most relevant features for classifying the user’s perceived loneliness levels.Three types of data used for classification task were indoor data, outdoor data and combined indoor-outdoor data. Indoor data consisted of indoor mobility data and statistical features from accelerometer data, outdoor data consisted of outdoor mobility data and other parameters such as speed recorded from sensors and course of a person whereas combined indoor-outdoor data had common mobility features from both indoor and outdoor data. We found that the machine-learning model based on XGBoost algorithm achieved the highest performance with accuracy between 90% and 98% for indoor, outdoor, and combined indoor-outdoor data. We also found that Lubben-scale based labelling of perceived loneliness works better for both indoor and outdoor data, whereas UCLA scale-based labelling works better with combined indoor-outdoor data.

## 1. Introduction and Motivation

Loneliness and social isolation are prevailing especially in the aging population. These two factors are considered as a major public health concern for the elderly people. Although loneliness is not a permanent condition, it can also be chronic in nature [1]. If not addressed, loneliness and social isolation could even lead to numerous health problems such as hypertension, diabetes, heart problems, mental disorders, etc. Moreover, loneliness feelings can be triggered by numerous factors, such as changes in the built environment, living conditions, life changes, mobility, or loss of strength or health, etc. [2,3,4].

Our previous work in [3] introduced the idea of a machine-learning based architecture for monitoring the levels of social isolation and/or perceived loneliness in the elderly with the help of wearable sensors. Through [3], hypothesis have been studied via proof-of-concept using the dummy data which was continuous in nature and resembling sensor data. Machine learning algorithms have been used to develop model to identify the relationship between sensor data and risk of loneliness. However, the previous study was based on dummy data. In this study we have used real time indoor and outdoor sensor data collected from participants to develop a model for identifying loneliness. Moreover, this study have also identified various indoor and outdoor mobility patterns which could be useful in identifying user’s behavioural context. Various wearable sensors from the market for the monitoring of loneliness, such as Pozyx system, Oura Ring, Pico MiniFinder, Moodmetric Ring, Withings ScanWatch, Imosi Smart Bracelet P11, Fitbit Luxe and Garmin Instinct were analyzed and assessed in [3] in terms of their attributes, energy consumption, obtrusiveness, and ease of data extraction and two sensor categories were identified as offering promising features in the context of loneliness and social-isolation classification and prediction: Ultra-WideBand (UWB) band Pozyx brand for the indoor location measurements and Global positioning System (GPS)-based Pico minifinder for the outdoor location measurements. In addition and considering the demography of the participants in our study (i.e., the elderly), it was necessary to select the two sensors as they offered the most convenient means for the elderly to report the current state of their mood at every point in time through the use of the push buttons. In order to analyze the data from these sensors, various machine-learning algorithms were also identified in our previous study [5]. The most used machine-learning algorithms in the literature for such studies are: the gradient boosting algorithms and the XGBoost which were used, for example, in the English Longitudinal Study of Ageing (ELSA) dataset to predict the loneliness [1]. Machine learning algorithm such as naive Bayes approaches have also been used, for example in [6] in order to develop elderly-based monitoring systems using accelerometer data to detect the movements of the elderly. Moreover, there are studies that have proposed various Internet of Things (IoT) platforms which rely on wearable sensors for elderly health care, for example in [2].

Connections between loneliness feelings and mobility patterns have been previously analyzed in [7] based on data collected via questionnaires; no sensor data were collected in [7]. The main findings were that people frequently visiting public spaces, such as community areas, sports facilities, parks or gardens, were less lonely than those with less frequent visits to such public places. Table 1 shows the comparison between the current study and the previous studies in the area of identifying loneliness. It can be seen from the Table 1 that only a very limited amount of research has so far used so far machine-learning methods on data related to mobility patterns to predict loneliness. The studies are based on using only questionnaire or neighbourhood data or use of public spaces and certain datasets based on mobility patterns for predicting loneliness whereas other studies which have focussed on using a machine learning approach use various health parameters for elderly monitoring. However, our study focuses on using the data from indoor and outdoor mobility-related sensors to identify the relationship between a user’s loneliness and their mobility.

The main goal of this study has been to find correlations (or features) relating the sensor-harnessed data to quantitative indicators or metrics which play an important role in identifying the loneliness or social isolation. For this purpose, two main loneliness metrics were identified based on our previous work in [5]: UCLA scores and Lubben scores are two such indicators which provide information about a person’s feelings of loneliness and social interactions. The UCLA (University of California, Los Angeles, CA, USA) loneliness scale is used as a measure of loneliness while the Lubben social network scale measures the social support received by family and friends further description of these scores is given in Section 3. As mentioned in [7], elderly people with more social interactions are generally feeling higher feelings of satisfaction as compared to those who have fewer social interactions and less social activity. Similarly, low-quality social relationships in older adults are strongly associated with feelings of loneliness as stated in [8] which examined the physical activity interventions with risk of loneliness in older adults. The research in [8] also used the UCLA (University of California, Los Angeles, CA, USA) loneliness scale and the Lubben scale to assess the loneliness and to categorize user’s social networking. In this paper we investigate the association of UCLA and Lubben scores with the indoor and outdoor mobility patterns of the user based on sensor-collected data.

The rest of the paper is organized as follows: research questions being addressed in this study are described in Section 2. Section 3 describes the process of data collection along with the sensors as well as the pre- and post- study interviews details. Section 4 provides a detailed description of the methodology used in this study. This sections describes the data cleaning, exploratory data analysis, feature extraction, feature selection, and the investigated machine-learning algorithms. Section 5 presents the result of the analysis for different machine-learning algorithms. Section 6 discusses the key findings of the study along with future directions for the work.

## 2. Research Questions

The objective of this study was been to explore relationships between the mobility patterns collected automatically via indoor and outdoor sensors and the metrics reflecting feeling of loneliness or social isolation of elderly. This study relies on using a machine-learning based approach on indoor and outdoor sensors data for classifying the loneliness levels of a person. This paper has also focused on identifying which type of data, indoor, outdoor or joint indoor-outdoor could better classify a person’s loneliness level. Moreover, this study has also contributed towards identifying various mobility-related features which plays an important role towards loneliness level classification.

Measurements were conducted with volunteers frequenting an elderly nursing house/community center in Tampere, Finland (see details in Section 3). The indoor and outdoor data were collected with two types of sensors: Pozyx as the indoor sensors and Pico minifinder as the outdoor sensors.

The four main research questions addressed by this study are:RQ1What kind of mobility patterns or features can be extracted from indoor and outdoor sensors in a manner relevant to loneliness and social isolation studies?RQ2How much are these mobility patterns correlated with loneliness metrics such as the UCLA and Lubben scores?RQ3How well can we classify the user by using machine-learning algorithms with a user’s indoor, outdoor, and combined indoor-outdoor mobility patterns and which of this data types gives the highest classification accuracy?RQ4To what extent can one predict the indoor mobility patterns of a user by training the machine-learning algorithm on the outdoor mobility patterns of the same user and vice versa (i.e., predicting/classifying outdoor from indoor data)?

Answers to these research questions are summarized in Section 6.

## 3. Description of Data Collection

This sections includes the details of the data collection process. The data were collected within a project run at Tampere University, called AISOLA [9]. The datasets created and collected for the technical study of AISOLA project and in this paper have been of two main types: (i) in-person surveys or questionnaires and (ii) sensor-based data with sensors carried by volunteers and automatically collecting positioning-related data, such as estimated speed and latitude and longitude coordinates.

The surveys were conducted twice: first in the so called pre-study stage (i.e., before the sensor devices were given to the participants) and the second one in the so-called post-study stage (i.e., after the devices were retrieved from the participants). In between the pre- and post-studies, the participants were given two sensors each to carry around (one for indoor and one for outdoor measurements). The sensors were given at successive times, not simultaneously, in order to avoid confusions: first, the indoor measurements were performed with the Pozyx sensors, and afterwards, the outdoor measurements were performed with the Pico Minifinder sensors.

The pre-study survey, conducted in person, was organized in order to get a baseline profile of the individual participants and their existing levels of social isolation and loneliness through the Lubben Social Network Scale-6 (LSNS-6) and the UCLA 3-Item Loneliness Scale, described in the following subsections; the post-study survey was also based on a in-person questionnaire collecting feedback regarding the devices and the participants’ physical/emotional wellbeing. In between the pre- and post-studies, data was collected via indoor and outdoor sensors by the volunteer participants.

The sensors were primarily utilized to collect the location data of the individual participants, while they were indoors or outdoors. Two separate sensor devices (with their own respective principles of operation) were utilized to collect the location data, in real-time and for the entire duration of the study. The reason to use different sensors indoors and outdoors was the fact that high accuracy is achievable with Global Positioning System (GPS)-based sensors outdoors and Ultra-Wideband (UWB)-based sensors indoors.

Pico Minifinders (https://minifinder.com/products/pico, accessed on 27 May 2022) sensors were used for monitoring/collecting the position and movements of each individual participant outdoors (using GPS). The monitored indoor spaces were the common gathering areas of the elderly in the communal nursing facility (which includes the gymnasium, cafeteria, activity center and the art center) and it was utilizing UWB technology from Pozyx (https://www.pozyx.io/technology/uwb-technology, accessed on 27 May 2022). The indoor sensors could also be utilized to document the broad emotional status of the participant, at any given point (while they are inside the monitored area). The next sub-sections give details regarding the demographic profile, various types of data collected, the nature of the collected data, and certain limitations of the sensor systems.

### 3.1. User Demographics

There were a total of six individual participants participating in all the steps of our research (pre-study surveys, sensor collection, and post-study surveys) and they were all female. However, we were able to gather sensor data from only five of them, as one of them was hospitalized during the study. All the participants were provided both the indoor and outdoor sensors. Most participants used a lanyard to carry the Pozyx tag and a key chain to carry around the Pico Minifinder. The participants were in the range of 68–87 years of age (with an average of 80.2 years), and were all mentally and physically capable to understand the requirements of the study and were fully involved.

### 3.2. Pre-Study Survey

The pre-study survey consisted of a combined questionnaire for the Lubben Social Network Scale-6 (LSNS-6) and the UCLA 3-Item Loneliness Scale.

The LSNS-6 is a tool used to document and quantify the social isolation among the older adults, by determining the number of contacts and the frequency of contact with their family members and friends, while also determining the perceived amount of social support received by the older adult from these sources [10]. The questionnaire has two sets of three similar questions (six questions in total), for friends and family members respectively; and each question has six options to choose from, with each option being assigned a score from 0 to 5 (total score of 30). The result of the LSNS-6 is the total sum of the individual scores of the selected option for each question. A single set of the LSNS-6 questions are as follows: “How many relatives do you see or hear from at least once a month? How many relatives do you feel at ease with that you can talk about private matters? How many relatives do you feel close to such that you could call on them for help?”. The same set of questions are repeated in the section of the questionnaire for friends. The options for response to each question are as follows: none; one; two; three or four; five through 8; nine or more and they are given weights ranging from 0 to 5, respectively (with none = 0 and nine or more = 5). A cumulative score of 12 and lower indicates that the individual is “at-risk” for experiencing social isolation. The LSNS-6 was selected among the two loneliness metrics of interest as it shows high internal consistency and consistent factor structure, and it is reliable across varying demographics and health characteristics, making it a more reliable tool to easily measure social isolation as compared to its counterparts.

The UCLA 3-Item Loneliness Scale is a simplified derivative of the Revised UCLA Loneliness Scale (R-UCLA test) which is renowned for its efficacy in quantifying and documenting loneliness in individuals, across multiple demographics and sample sizes [11]. The UCLA 3-Item Loneliness Scale condenses the essence of the 20-point questionnaire of the R-UCLA test into 3 broad stroke questions which make conducting the survey easier and less time consuming, especially across a large population. The UCLA 3-Item Loneliness Scale also reduced the number of response categories, with each option carrying a different weights. Similarly to the R-UCLA test, all the responses of the individuals in UCLA-3 item test are summed and the higher overall scores are indicating a correspondingly greater degree of loneliness experienced. The UCLA 3-Item Loneliness test entails the following questions: “How often do you feel that you lack companionship? How often do you feel left out? How often do you feel isolated from others?”. The response categories are: “Hardly ever (1 point); Some of the time (2 points); Often (3 points)”. The UCLA 3-Item Loneliness test has demonstrated itself to be robust and reliable in different interview modalities (self-administered in person and over telephone). It could be used as a standalone measure for loneliness, or it could also be embedded within the R-UCLA test, for cross evaluation purposes [12]. However, for the purpose of our study, the classifier information provided by the UCLA 3-Item Loneliness Scale was found to be more valuable individually rather than in combination with R-UCLA test, as it provides all the nuanced information in useful broad strokes.

### 3.3. Post-Study Survey

The post study survey collected information from the participants of the study regarding their physical and emotional state during the study, device usability and ease of maintenance, general information regarding their hobbies, as well as some general feedback (which we could use to improve our next studies and practices with the next sample size). On device usability, the participants ensured that they carried the indoor and outdoor devices with them at all times as mentioned during the information session held with them, while giving out the devices. The caregivers at the elderly home also played a role in reminding the participants to carry the devices with them. In addition, there were regularly scheduled weekly calls with the participants to follow up on device usage as this was very critical to obtaining quality results.

The participants mostly interacted with the indoor device, whose button they had to always push to reflect their current mood (neutral, happy or sad). On the other hand, there was less interaction with the outdoor device as the participants only had to carry it around to track their locations at all times. To ensure the quality and appropriateness of the data gathered, the indoor and outdoor were collected continuously over a period of one month. The indoor data were collected at a frequency of 8 h per day and outdoor data, 24 h daily. With this, an adequate amount of indoor and outdoor data were collected to build the machine learning models for predicting loneliness in the elderly.

The post study interview questionnaire can be found in Appendix A. The questionnaire could be utilized to cross examine and verify the usage patterns of the sensor devices and to provide context to the collected sensor data. All the questionnaires provided to the participants were translated to the native language of the participants (i.e., Finnish), in order to provide them unrestricted clarity and understanding. Researchers were also present while distributing the devices and conducting the surveys, to explain the objectives of the study and to clarify any questions that the participants had regarding the devices, collected data and the study outcome.

### 3.4. Indoor Data Collection

The indoor data were collected by utilizing the indoor positioning systems operating based on UWB, which is a wireless radio technology having a wide spectrum of 500 MHz and offering position accuracies indoors of up to 10 cm [13]. Additionally, the system has some (limited) provisions to capture the emotional state of the participant being monitored, based on optional manual feedback from the participants. The participant can manually interact with their Pozyx sensor device to reflect their current mood. The system also has the capability to create virtual zones in the map to be monitored and the zonal information is also recorded when a participant carrying the sensor device walks into the region marked as a zone.

The hardware aspects of the Pozyx system comprise an interactive tag (worn by each of the participants being monitored and automatically transmits the positioning signals and the device state information to the anchors), the anchors (which are strategically fitted across the built environment to detect signals from the interactive tags, process the data, and make it available to the gateway), and the gateway (which is essentially the CPU for the system, capable of processing the information from the tags and anchors, and gives the researchers access to real-time positional and device related data).

The hardware components which are utilized in the existing Pozyx setup (https://www.pozyx.io/products/hardware/kits, accessed on 27 May 2022) are shown in Figure 1.

The hardware setup is finalized once all the anchors are fitted, and their locations are calibrated with respect to each other and the virtually defined Cartesian origin.

Figure 2 shows the regions monitored for the current implementation of the study, and they included: the cafeteria, the gymnasium, the activity center (’päivätoiminta’ on the map) and the arts center.

Each anchor requires 5 W of power to function properly and multiple anchors can be daisy chained to form a single branch of anchors. The anchors are powered by utilizing Power over Ethernet (PoE) technology and they are connected to a PoE switch, to provide the anchors power and to enable the connection of large number of anchor branches to the gateway (to minimize the cable and LAN port consumption). Figure 3 shows the anchors physically mounted in the various sections of the elder care premises.

The interactive tags carried by the participant transmit their Cartesian coordinates along with the device status (representing the participant mood in this study). The device state can be changed easily (manually) with the push of the only button on the tag, and the tag can have three states (which can be cycled through by pushing the button). The three moods identified by the study are ‘Neutral, Happy and Sad’. The tag has an LED indicator which provides visual feedback to the participant regarding the state of the tag: green was associated with neutral; blue was associated with happy; and red was associated with sad. The state and its corresponding LED color was labelled at the back of the tag, for the convenience of the participants as a reminder. The interactive tags also have inbuilt accelerometers which provide acceleration data across *X*, *Y* and *Z* coordinates. This could provide valuable information regarding the speed of movements which could help build the mobility profile of the participants. The position and accelerometer values of the individual tags are updated at a high rate of 1.6 Hz to the anchors, to ensure good accuracy in the collected data.

The gateway processes all of the information and has provisions to provide analytics of the collected data to visualize heatmaps, anchor-tag participation, tag tracing, etc. However, for the requirements of the study, we utilized the capability of the gateway to publish the real-time location, emotional, and acceleration data of all the participant tags to the cloud. By utilizing the MQTT (Message Queuing Telemetry Transport) protocol, which is a lightweight network protocol used in locations with resource constraints, we can access the positioning data uploaded to the cloud and the newest entries are recorded as soon as they are made. The real time data entries are all in JSON (Javascript Object Notation) format and are collected into .txt files on a day-to-day basis (data collection for a particular day begins at 09.00 h and ends at 17.00 h). There is a dedicated computer which automatically begins and terminates the data collection for the indoor data. Each individual entry is a combination of JSON objects and arrays representing the information of a single tag, which is obtained from the Pozyx cloud through an MQTT subscribe protocol. A sample JSON message is shown in Figure 4, and the essential information contained in its message structure is shown in Table 2.

The limitations for this particular positioning system are minimal; they are as follows: only indoor location data with a local coordinate system could be collected and the participant must physically carry the tag to all places within the built environment. However, the tags are very compact (50 × 42 × 15 mm) and lightweight (21 g) and do not require any charging for the battery for the duration of the study, thus they are very easy to maintain.

### 3.5. Outdoor Data Collection

The outdoor data were monitored and collected using the Pico MiniFinder, which is based on the Global Positioning System (GPS) [14]. The Pico MiniFinder facilitates real-time GPS tracking of persons and assets. It has a pre-installed SIM card through which the positional data (GPS coordinates) of the device is captured and transmitted to the cloud. The SIM card also ensures that the device can be polled at any time to get the real-time position. The GSM operating frequency of the Pico MiniFinder is in the range of 850/1900 MHz 900/1800 MHz. The GPS sensitivity is about −158 dB and the active fix time against GPS is 1 s. Configuration of the Pico MiniFinder is possible through the MiniFinder mobile app or by logging to the device page on the MiniFinder web portal (https://go.minifinder.com/, accessed on 27 May 2022). The Pico MiniFinder is supported by a server-based software (MiniFinder GO), which monitors the GPS trackers. The captured real-time location data are then processed, stored, and presented on a map.

The hardware components of the outdoor monitoring system consist of an outdoor tag, which is very light (35 g) and carried around by the participants. It can be attached to keyholders or worn around your neck with a lanyard or inside a handbag or pockets, while going out. The main function of the outdoor sensor is to to monitor the location of the participants when they are doing activities outside the elder care facility. The outdoor tag consists of components such as a panic button, microphone, speaker, charging contact pins, call button etc. as shown in Figure 5. It requires very minimal interaction with the participants and they only have to charge (usually once in two days) it when the battery is low. A blue light blinks on the device when it is charging and then goes off when it is fully charged. For every outdoor tag (via the MiniFinder GO app), there are four different colours (Green, Orange, Red, and White) that indicate the status of the GPS unit. This is illustrated in Figure 6 and Table 3.

In addition, different alarms such as Geofence, panic alarm, fall alarm and low battery alarm can be configured on the MiniFinder GO app to become alerted when any of the situation arises. A notification for a low battery alarm is shown in Figure 7. The low battery alarm is triggered when the battery percentage drops below 20%. This alerts the device carrier to charge the device battery. Although as a rule, the participants were directed to charge their device batteries once every two days. The rate at which the battery power of the device is consumed is a function of the position update interval i.e., the time interval between which the location of the device is tracked. For the Pico MiniFinder, the position interval varies between 30 s and 3 min. The longer the position update interval, the less the battery consumption. For this study, the position update was set at the maximum of 3 min to conserve battery usage as there is usually less mobility by the participants.

Following the tracking of the outdoor devices, the real-time location data of the outdoor devices are processed and stored in the cloud. They are presented in NMEA format and downloaded as .txt files on a daily basis as the data collection runs for 24 h. The location data collected consists of the timestamp, longitude, latitude etc. The location data are downloaded from the MiniFinder web portal. Figure 8 shows a snippet of a recorded entry for an outdoor tag, and the information available from the National Marine Electronics Association (NMEA) data format (see below):


**$GPRMC,200637,A,6128.2257,N,02346.8965,E,0.54,126,280522,,**


**Figure 8 sensors-22-04946-f008:**
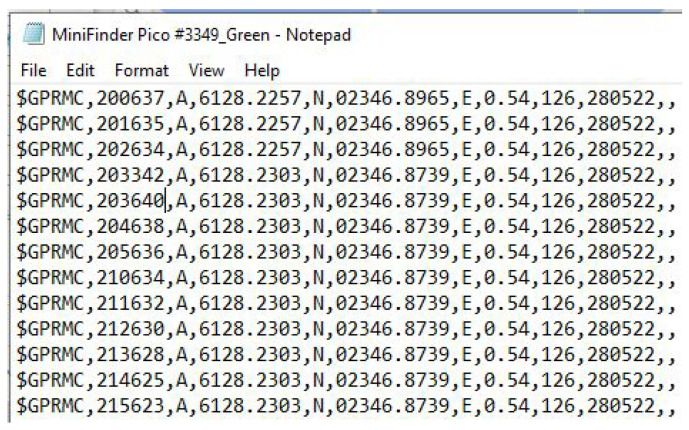
NMEA message showing outdoor tag data.

The data are presented in as GPS Mobile Recommended Minimum Configuration (RMC) sentence ($GPRMC) sentences, shortly known also as the Recommended minimum specific GPS/Transmit data. The structure of the recorded entry, detailing the first line of the entry in Figure 8 is shown and explained in Table 4.

A limitation of the Pico MiniFinder is the constant need to charge the batteries. Another limitation is associated with the fact that it is dependent on GSM networks, which means that location data is not transmitted when there is no GSM coverage.

## 4. Machine-Learning-Based Methodology and Analysis

Machine-learning algorithms are used in various classification, prediction, clustering applications [15,16,17,18]. The step-by-step methodology followed before feeding the data to machine-learning algorithms is highlighted in Figure 9.

The machine-learning algorithms are highly dependent on the available data. The cleaner the data are, the more accurate the model and its evaluation matrix are. Prior to feeding the data into our machine-learning models, it is also important to observe the distribution, patterns, trends, or features in the data through statistical analysis or visual representations, such as scatter plots, bar plots, histograms etc. This process helps in identifying the association and main characteristics in the data. This approach of performing initial investigations is known as exploratory data analysis. Exploratory data analysis is followed by a feature-engineering process. Feature engineering includes feature extraction, feature selection, or feature elimination methods. For example, the feature extraction is required to identify the time-domain and frequency-domain features from the continuous data. Similarly, feature selection methods such as wrapper-based methods and filter-based methods can help to select the most relevant features. Based on the correlation between the variables, related or redundant features can be eliminated from the data. This feature engineered data are then given to the machine-learning algorithm to generate the model. The indoor sensor data collected for this study consists of various attributes such as timestamp, accelerometer data of person, *x*-*y*-*z* coordinates of each person, cluster or zone in which the user was present, and the emotional state as manually defined by each user (happy, sad, or neutral). The collected outdoor sensor data consists of attributes such as time, date, speed, latitude and longitude, the course of the person which is direction of progress of a person, between two points, with respect to the surface of the earth. Data related to person’s perceived social support and feeling of loneliness as well as social isolation was also collected using the UCLA loneliness scale and Lubben’s social network scale based on the pre-study survey. Before training the model using the indoor or outdoor data or both, it is important to pre-process the data, to extract meaningful mobility patterns and to understand the characteristics and associations between these patterns. We explored research questions RQ1 and RQ2 mentioned in Section 2 through the initial analysis.

### 4.1. Data Pre-Processing

A step-by-step process based on Figure 9 was followed to obtain the meaningful information from the raw indoor and outdoor sensor data. First step to be carried out in this process is data pre-processing. Indoor and outdoor data collected from the sensor required two steps to be carried out for data cleaning as a part of data pre-processing:Converting date and time information into “YYYY-MM-DD HH:MM:SS” format.Extracting relevant attributes from the sensor data. This includes eliminating the metadata from the raw sensor data such as error information, version information, message identifier (ID) etc.

### 4.2. Exploratory Data Analysis

To understand the behavioural context of a person, certain parameters need to be extracted from these indoor and outdoor data which will represent person’s mobility patterns. For the available data, we extracted several parameters such as estimated speed, distance travelled, frequently visited spots, time spent at those spots, etc. This subsection provides the description of the methodology for identifying the clusters.By means of visual representation, this subsection also provides an overview of the average time spent by users in clusters.

Identifying clusters or frequently visited placesFor indoor data, the clusters or frequently visited places indoor were recorded by the sensors. For outdoor data, the clusters or frequently visited hotspot information were not recorded by the sensors. Outdoor sensors recorded only the information about the latitude and longitude of a person. We utilized this information to identify ourselves the clusters or frequently visited places outdoors. We used k-means clustering algorithms to identify each user’s frequently visited areas. These clusters or frequently visited places were recognized better after plotting the data on the maps. Figure 10 shows the outdoor clusters or frequently visited places identified for two users. Each color represents the different clusters. The black circles denote the cluster boundaries. A user can have several clusters; in the examples shown in Figure 10, both users have had three main clusters. The underlying map (also used in our analysis) is not shown here in order to preserve the users’ privacy.Based on the information obtained from the mobility patters, we performed exploratory data analysis to explore the behavioural context of the users. We summarized the information obtained from this analysis in the form of bar plots. The bar plot represented in Figure 11 tell about percentage of time spent by users in indoor and outdoor clusters.Average percentage of time spent by users in indoor and outdoor clustersTo obtain the average percentage of time spent by users, we used the cluster information and the days and times the user visited those clusters. For analyzing and visual representation of the average time spent by the users, indoor data were available from five users whereas outdoor data were obtained from six users. Users IDs are from 1 to 7, with some users missing, as initially 7 participants were enrolled to the trial, but one participant was retrieved before the process started, and a second participant was retrieved (due to health reasons) after the outdoor data was collected, but before the collection of the indoor data. So, for indoor data from user 1, 2, 3, 4, 7 were used and outdoor data from user 1, 2, 3, 4, 6, 7 were used. The average percentage of time spent by users in different places indoors and outdoors is represented by bar plot as shown in Figure 10.For indoor clusters, all five users—whose indoor data were being collected—visited up to four places frequently, marked by Place1, Place2, Place3, Place4 (not necessarily the same places for all users, therefore Place1 of user1 may be at a different location from Place2 of user2). They visited these places on certain days. The bar plot in Figure 11 shows that user1 and user2 visited only two places and spent most of the time almost 98% to 100% at one place on all the days. However, user3, user4, and user7 visited multiple places and spent significant amount of time in more than two places. For example, user3 spent about 78% of her time at one place, around 20% at a second place, and about 2% at a third place indoors. Similarly, user4 and user7 are also spending time in different places. User4 seems to spend more than 80% of time at one place but she is also interested in visiting other places, while user4 is spending around 5% to 7% at other places as well. Similar behaviour is observed in the mobility patterns of user7 as well. On one hand, it can be observed that user1 and user2 have less mobility indoors than the other three users, and, therefore, less physical activity. On other hand user3, user4 and user7 have better mobility as compared to the other two users and therefore better activity indoors.From the outdoor data and the computed clusters, we observed that all the users have been visiting to multiple places as per their interest and so overall, there are quite many clusters identified outdoors. Some of these clusters are common between the users. The bar plot in Figure 10 for time spent in outdoor clusters shows that user1, user2 have been spending most of the time, about 90% to 100% at one place on most of the days. They have been visiting to other places but not more than 2%. For user1, user2, percentage of time spent in other places is quite less. So, these users might have less social engagement outdoors. Similar behaviour is observed for user4 as well. User4 is also spending most of the time, around 98% in one cluster and not more than 5% in other clusters. However, if we observe outdoor behaviour of user3, user6, user7 then it is slightly different from the behaviour of user1, user2, user4. Although, user3 is spending around 85% in one cluster but user3 is also visiting and spending 5% to 15% of time in other cluster as well. User6 and user7 have also been visiting to multiple clusters and spending significant time in different clusters. This shows that there is a chance of user3, user6, user7 having better social engagement and mobility outdoors.We also compared information obtained from the indoor and outdoor mobility patterns with the Lubben score of the users. Lubben score obtained from 6-item social network scale tells about the social engagement of the user. The higher the score, the better the social engagement of the participant is. We observed that user1 and user2, who have less mobility indoors and outdoors than the other three users, also have lower Lubben scores. User3, who has better mobility patterns indoors and outdoors than user1 and user2, has good Lubben score. However, user4, who has better indoor mobility than user1 and user2, but lower outdoor mobility, has intermediate Lubben score values. We also observed that for user6 and user7, who have better indoor and outdoor mobility that user1 and user2, have very low Lubben score values. From the above information, it can be inferred that the mobility patterns can be associated to some extent with the social engagement of the person and hence the feeling of loneliness. Also, the average time spent in multiple places indoors and outdoors can also be used as one of the important characteristic to understand the mobility of users and this can help to understand the user’s behaviour such as social engagement, feeling of loneliness, etc. For a better understanding of the findings from the exploratory data analysis we have performed the classification task on the users’ mobility pattern using machine learning algorithms, the results of which are presented in Section 5.

### 4.3. Feature Extraction and Feature Selection

Feature-extraction methods refer to the process of extracting relevant numerical values from the raw data. For large amount of continuous data collected from the sensors, it is required to sample the dataset into small window size and to extract the relevant features out of it. The feature extraction is followed by the feature selection methods. Feature-selection methods can help to improve the performance of the model by selecting the most relevant features of the model. This subsection provides the information about the extracted features and the selected features from indoor and outdoor data for the classification task.

Indoor data feature extraction and selectionThe raw data collected from the indoor sensors consist of attributes containing information about the timestamp, user ID, user’s positional coordinates, three axis accelerometer data, cluster ID, cluster name, emotional state of person, additional metadata related to error information, latency, communication success rate, etc. After initial pre-processing on the various attributes, raw data containing accelerometer values, user’s positional coordinates, cluster information, and emotional state were considered further for feature extraction. User’s position coordinates were used to calculate the distance and the speed of the user. Raw accelerometer values in three axis were transformed to extract the statistical features from them.We used a windowing technique with a window size of 10 s and we applied feature-extraction methods on that window size. Various statistical features such as mean, median, standard deviation, kurtosis, skewness, number of peaks, energy of signal, signals magnitude area, and average resultant acceleration were extracted from the three axis accelerometer data. The average value for these transformed statistical features were taken over the considered window period. Features such as traveled distance and speed, calculated from the position coordinates were also averaged over the window period. For categorical variables such as cluster ID and emotional state of person, the most frequent value in that window period was taken. This transformed feature vector consisting of statistical feature, average distance, average speed, cluster information from all windows was formed by appending the different individual features. This transformed dataset (or feature vector) was given as input to the machine learning algorithms for further analysis.Outdoor data feature extraction and selectionThe raw data collected from outdoor sensor consist of attributes containing information about the message header, UTC (Coordinated Universal Time) time, status tag to indicate if the data is valid or not, speed over ground in knots, latitude and longitude information, course over ground, north/south indicator, east/west indicator. In outdoor data, the information about the frequently visited places or cluster was not automatically recorded by the sensors, unlike in the indoor measurements. As mentioned in previous subsection, we extracted the cluster information from the latitude and longitude attributes using k-means clustering algorithm. To extract the features such as distance and the estimated speed, latitude and longitude information was converted into Cartesian coordinate system. The *x*, *y*, *z* coordinate information were further used to calculate the distance and the speed of the user.For outdoor data, a smaller window size was taken to obtain the transformed dataset since the time difference between the two consecutive values was 10 min. We used a similar methodology as used for indoor data in order to obtain the complete feature vector. Here, we used the window size of 40 min. However, due to the difference of 10 min. between consecutive values a particular window consisted of 4 values. We calculated the average distance, average estimated speed, speed (converted from knots to meter/second) and average course over a window period. For a categorical attribute such as the cluster ID, we used the most frequent value in the window. The values from multiple windows were appended and the transformed outdoor dataset (or complete feature vector) was further given as input to the machine-learning algorithm.Indoor and outdoor feature fusionFeature fusion combines the features obtained from the different sources into a single feature set. To obtain the loneliness prediction accuracy for the combined indoor and outdoor mobility patterns, we applied a feature level fusion. The main challenge for indoor and outdoor feature level fusion was to take the features from the same user on same days. Only the features common to both the indoor and outdoor data were selected. The selected features were: the estimated speed, distance, and indoor cluster ID from the indoor data and the estimated speed, speed over ground, distance, and the outdoor cluster ID from the outdoor data. For selecting the relevant features, we calculated the correlation matrix.It was observed from the correlation matrix of all users that the indoor distance and indoor estimated speed were highly correlated. Estimated speed and speed over grounds from outdoor data were also correlated. Similar behaviour is shown from the example correlation matrix for two users, in Figure 12, where the indoor distance and estimated speed indoors are highly correlated with a correlation value of more than 90%. Also outdoor speed over ground and outdoor estimated speed were also positively correlated.The combined dataset from the indoor and outdoor data was given as input to the machine-learning analysis and we found that the best performance was achieved with the indoor distance used as a feature extracted from indoors and with the speed over ground used as a feature extracted from outdoors.

### 4.4. Machine Learning Based Analysis

In here, we focus on using machine learning algorithms for the classification of loneliness risk of elderly using the indoor and outdoor sensor data. This subsection describe in detail about the machine learning methods being used. In the context of this study we performed three types of classification tasks.

Classification of users (user1 to user4 and user6 to user7) based on their mobility patterns and other extracted features.Classification of users into three labeled classes, having high risk of loneliness, medium risk of loneliness, and low risk of loneliness with labels assigned based on the UCLA score of the user as mentioned in Table 5.Classification of users into three labeled classes, having high risk of loneliness, medium risk of loneliness, and low risk of loneliness with labels assigned based on the Lubben score of the user as mentioned in Table 5.

These three types of classification were performed on indoor data, outdoor data, and an indoor/outdoor combined feature set. In order to identify the user’s risk of loneliness, different types of data were used. The data obtained from indoor and outdoor sensors were continuous in nature, having a floating point data type. Additionally, the variables representing the clusters and user’s emotional state were categorical in nature. Input data consisted of features from indoor, outdoor and combined indoor-outdoor data. Features used for loneliness classification using indoor data were statistical features such as mean, median, standard deviation, kurtosis, skewness, number of peaks, energy of signal, signals magnitude area, and average resultant acceleration from accelerometer data, mobility features such as traveled distance and speed, cluster ID and emotional state of person refer Section 4.3. For outdoor data, mobility features were used such as average distance, average estimated speed, speed in knots (obtained from sensors), course, cluster ID refer Section 4.3. For combined indoor-outdoor data, mobility features common to both indoor and outdoor data were used such as the estimated speed, distance, and indoor cluster ID from the indoor data and the estimated speed, speed over ground, distance, and the outdoor cluster ID from the outdoor data, refer Section 4.3. Output data for loneliness classification is represented in Table 5. Various machine-learning algorithms can be used to analyze this heterogeneous type of data. However, from the study conducted in [5] to analyze the sensor data, it was observed that the algorithms mostly used for analysing continuous data from sensor are neural networks, support vector machines [19], random forest [20] and ensemble algorithms. Additionally, in our previous work on managing perceived loneliness in [3] we presented a hypothesis using machine-learning algorithms to identify the relationship between sensor collected data and loneliness levels. The algorithms we used in [3] were logistic regression, support vector machines, random forest. These algorithms were applied on the continuous floating point data and the categorical data. Although we performed binary classification with two loneliness levels but logistic regression, support vector machine, random forest were able to classify these loneliness levels very well. Since the data we used in this study is also floating point and categorical in nature, so we used same set of algorithms for the context of this study as well. We applied logistic regression, support vector machines, random forest for classifying the loneliness levels among elderly. We also included boosting algorithm, XGBoost [21] in our study since XGBoost finds the best tree based model and also works well with sparse and missing values.

Random forest is a type of ensemble algorithm using bagging technique with base estimator as decision trees. It has certain parameters such as nestimators, criterion, max features, max depth, min sample split, min samples leaf, max leaf nodes etc. For all the three classification task we have kept the default values for the hyperparameters except nestimators, which tells about the number of trees. Similarly XGBoost is also a type of ensemble algorithm which uses gradient boosting technique of combining weak learners to form strong learners. Moreover, it also works well for unbalanced datasets and in this study the classes were unbalanced for indoor data. Hyperparameters in case of XGBoost were set to default values. Support vector machines on the other hand perform the classification by dividing the classes through hyperplane. One of the important parameters for support vector machines is the kernel which helps in determining the shape of the hyperplane and decision boundary. In this study we have used the rbf kernel. The input data mentioned above were given to these algorithms to perform classification into three levels of loneliness.

For the classification problems, the most commonly used evaluation criteria as stated by [5] are accuracy, precision, sensitivity, and specificity. In here, we focused on two evaluation metric, namely accuracy and confusion matrix, which are the most encountered in research publications relying on machine learning and user data. The accuracy metric tells the correctly predicted observations from the total number of observations. Confusion matrix, tells about the model performance based on the test data for which the actual values are known. The confusion matrix is a good visual representation of what percentage of classes are correctly classified or mis-classified by a certain algorithm. Its diagonal shows the classification accuracies in the correct class, while the values out-of-the-diagonal show the mis-classification percentages. In our results, we normalized everything to 100%, in such a way that sum over rows or over columns in the confusion matrix always gives 100%. Visually, a darker color in the confusion matrix show higher percentages and a lighter color show lower percentages; a good estimator has dark colors along the diagonal of the confusion matrix and light colors outside the diagonal. We have used these two metrics (accuracy and confusion matrix) in order to evaluate the followings:How well the algorithm is able to identify the users based on the selected features from sensor data?How well the algorithm is able to classify the users into three defined loneliness levels (using the UCLA and Lubben loneliness metrics)?

Hence, to achieve a wider perspective on the relationship of loneliness levels to the mobility patterns, the extracted mobility features were given to tree-based ensemble algorithms such as XGBoost, random forest and to a support vector machine classifier and the algorithm performance was evaluated using an accuracy and confusion matrix. The results of the classification are presented in the next section.

## 5. Classification Results

In the previous section, we identified different features that can be used to classify the loneliness levels of a person. We also explored the correlation of a user’s mobility pattern with their behaviour. In this section, we provide a precise description of the result of applying machine-learning algorithm for the classification of user’s loneliness levels based on their indoor, outdoor and combined indoor and outdoor data. This section is further subdivided into the three subsection, the first subsection describes the result of classification using indoor data features, the second subsection describes the result of classification using outdoor data features, and the last subsection describes the result of classification using combined features from indoor and outdoor data. A comparative analysis on the results of classification using different data is also presented in this section. Results of different machine-learning algorithms for the classification of loneliness levels using indoor and outdoor data are presented in Table 6.

### 5.1. Indoor Data Analysis

The raw indoor data collected from the sensors were given to the feature extraction engine and for classifying the loneliness levels. We assigned three labels based on the score of user’s UCLA and Lubben scores. The labels symbolizes the output data. Details of input data and algorithms used are presented in Section 4.4. The results summarized in Table 6 shows that XGBoost have shown better accuracy in predicting the user as well as their loneliness levels. Figure 13 shows the confusion matrix for the XGBoost algorithm for all three types of classification. It can be seen from the confusion matrix for user classification that more than 90% of times user have been correctly classified based on their mobility patterns. Similarly, for UCLA and Lubben based classification, user’s loneliness levels have been predicted correctly more than 94% of time. Although the label distribution in case of indoor data was highly unbalanced but support vector machines and XGBoost were able to predict the minority classes also well. In the case of the UCLA based classification, the minority class label 2 indicating user’s having high level of loneliness is also correctly predicted in most of the cases. We identified that the most relevant mobility patterns for classification were cluster identity (ID) or frequently visited places, emotional state of a person, average distance travelled and average speed of a person. We found that for indoor data classification Lubben score based labelling provides better results as compared to UCLA based labelling, which means lubben score based labelling could better predict the risk of loneliness for indoor data. This also supports our findings from exploratory data analysis in Section 4.2 that user’s having more mobility/activity might have lesser chance of feeling lonely. The results also highlights the perspective that the user’s social engagement is associated with the mobility patterns as lubben scores explains more about the social engagement of a person.

### 5.2. Outdoor Data Analysis

The raw outdoor data collected from the sensors were given to feature extraction engine to extract the mobility related features such as outdoor estimated speed, distance travelled, frequently clusters visited, etc. We used two features obtained from sensors, they are speed and course over ground. From the other features available from sensors such as latitude and longitude we converted them into cartesian coordinates and derived features such as distance travelled. Other attributes such as datetime, north-south indicator, east-west indictaor, valid or invalid entry tag were mostly metadata. The outdoor data were available for six users. For classifying the user, the user ID was used as labels. For classifying the loneliness levels, we assigned three labels based on the score of user’s UCLA and Lubben score. The results summarized in Table 6 shows that XGBoost have shown better accuracy in predicting the user as well as their loneliness levels. Figure 14 shows the confusion matrix for the XGBoost algorithm for all three types of classification. It can be observed from the confusion matrix that in more than 85% of times the outdoor mobility patterns could classify the user correctly. The loneliness levels are also identified correctly for by UCLA and Lubben based labelling. However, the UCLA based labelling could predict the classes correctly between 89% to 94% whereas Lubben based labelling could correctly predict the classes between 91% to 97%. The most relevant features that could predict the user’s behaviour were the course of user, cluster, speed over ground and estimated outdoor speed.For outdoor data, Lubben score could predict better the feeling of loneliness. The results from outdoor data analysis supports our findings from exploratory data analysis in Section 4.2 that user’s having more mobility/activity might have lesser chance of feeling lonely and better social engagement.

### 5.3. Indoor-to-Outdoor Correlations

To identify the correlation between indoor and outdoor data, we first tried to classify user and their loneliness levels by combining the indoor and outdoor mobility features. We combined features such as indoor and outdoor cluster information, indoor and outdoor distance travelled and estimated speed, outdoor speed over ground. Both indoor and outdoor data were present for five users. Based on the feature relevance information obtained from the indoor and outdoor data based classification, we selected indoor and outdoor cluster ID, distance travelled indoor and outdoor, outdoor speed over ground as features for the classification. The result of classification is summarized in Table 6. From the results it can be seen that XGBoost showed better accuracy in predicting the user as well as their loneliness levels. Figure 15 shows the percentage of classes correctly classified. For user classification, except for user 1 all other users have been classified correctly for more than 93%. For loneliness classification, the loneliness levels are classified correctly for the users between 97% to 99% for both UCLA and Lubben-based classification.However, for combined classification, UCLA based labelling have showed better results, which means risk of loneliness is better predicted using UCLA scores. So, it can be inferred from the results than with combined indoor-outdoor data, social isolation is better highlighted as UCLA scores explains more about social isolation.

We also tried predicting the indoor and outdoor mobility patterns by training the machine learning models on either indoor or outdoor data and vice versa. To explore this possibility we trained XGBoost regressor with the indoor distance travelled, indoor estimated speed of one user. We tried predicting outdoor distance travelled and outdoor speed of the user. We found that for all the user’s data, the root mean square error values and the mean absolute error value were good, but the r-squared value which tells about the model fit was negative. This showed that the model was leading to worst fit than horizontal line. We also explored whether we can identify the user by training the model using either indoor or outdoor behaviour. We trained the model using indoor distance travelled, indoor estimated speed of all the users and tried predicting the user using outdoor distance travelled and outdoor speed. The accuracy of classification was 62% with the XGBoost classifier. We tried predicting the user by giving outdoor data to the model. The result showed that model could not identify user1 and user2 by their outdoor data. However, model could predict user3 by 75.4% and user4 and user7 by 9.71% and 2.3% respectively. We also trained XGBoost classifier on the outdoor data so that we could predict the user by using indoor data but the classification accuracy was very low. So, we found that using machine learning algorithms on indoor and outdoor mobility features, individually and combined could predict the user and their level of loneliness with good accuracy. However, predicting the user using training on either indoor or outdoor mobility pattern may not be possible using just two features speed and distance. It might be possible to identify significant relationship between the indoor and outdoor mobility patterns by incorporating more features and data.

## 6. Discussion

In this paper, we examined various machine learning algorithms to identify the user’s level of loneliness using a Pozyx sensor for indoor data and Pico Minifinder for outdoor data. We identified different indoor and outdoor mobility patterns that can be used as features to identify the loneliness levels and user’s characteristics. This study also explored the association between the user’s mobility patterns and their UCLA, Lubben scores. We also studied, if it is possible to predict either outdoor or indoor mobility patterns by training the model using either of indoor and outdoor data.

Exploratory data analysis on the indoor and outdoor data revealed that average time spent indoors and outdoors can be considered as an important characteristic for understanding a user’s mobility and these characteristics have some positive correlation with the Lubben score. We also found that to identify user’s characteristics or loneliness levels, indoor and outdoor distance travelled, indoor and outdoor estimated speed, frequently visited indoor and outdoor places, statistical features from accelerometers are some the mobility patterns which can be used as features for machine learning models. Machine-learning algorithms such as support vector machines, random forest and XGBoost were used for classification, and a evaluation criteria such as accuracy and confusion matrix was employed. Results from the machine learning study revealed that XGBoost performed better among all the other algorithms for classifying the user and their loneliness levels. XGBoost performed well with indoor, outdoor and combined indoor-outdoor mobility patterns with its accuracy range between 90% and 98%. For indoor data, support vector machines and XGBoost both performed well in the case of classifying the minority class labels. Moreover, we found that for indoor and outdoor data Lubben based labelling showed better results whereas for indoor-outdoor combined data UCLA based labelling showed better results.So, it can also be inferred that indoor and outdoor data highlights the social engagement of a person whereas combined indoor-outdoor data highlights more on the social engagement of the person. Through this study we also tried identifying significant relationship between the indoor and outdoor mobility patterns by training models on either of indoor or outdoor mobility patterns and predicting the patterns based on the other. However, we found that the possibility to predict indoor or outdoor user behaviour using the other behaviour is low with only indoor/outdoor distance travelled and indoor/outdoor estimated speed. However, it might be possible to identify more mobility patterns which could help in predicting one behaviour from other with good accuracy.

This study showed that it is possible to predict user classes based on metrics related to loneliness and social isolation with significant accuracy using indoor, outdoor, or combined indoor-outdoor sensor data and machine-learning algorithms. However, there is an as-yet unexplored-yet possibility of increasing the classification accuracy if data from more users were available. This could also help with developing more generalized models. Another open research issue is to further explore the mobility-specific information with additional sensors (e.g., accelerometers, gyroscopes, pedometers, etc.) to be able to establish a better association between the indoor and outdoor data belonging to the same user.

## Figures and Tables

**Figure 1 sensors-22-04946-f001:**
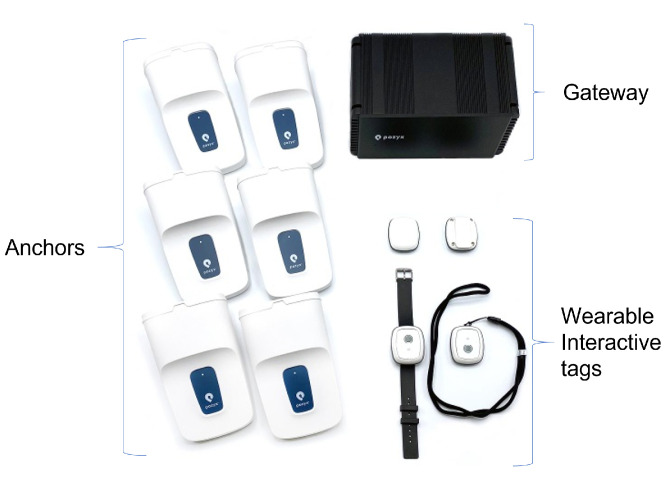
Pozyx Hardware devices.

**Figure 2 sensors-22-04946-f002:**
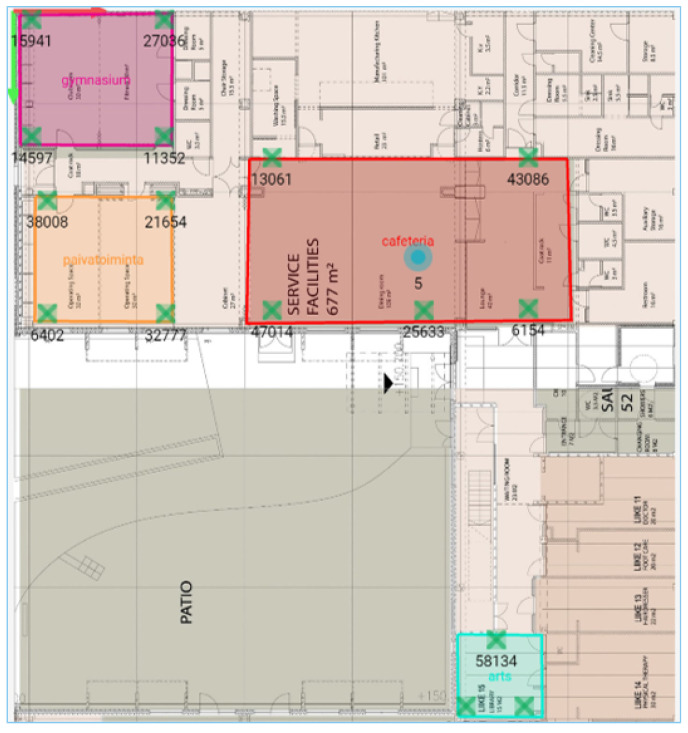
Virtual layout of the monitored environment.

**Figure 3 sensors-22-04946-f003:**
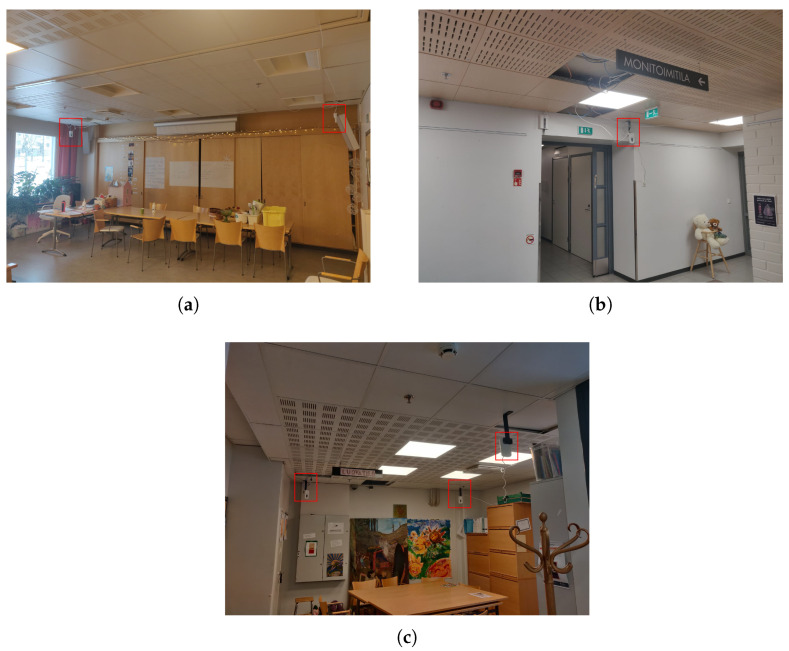
Anchors physically mounted in the elder care premises. (**a**) Activity Center; (**b**) Gymnasium; (**c**) Arts and craft center.

**Figure 4 sensors-22-04946-f004:**
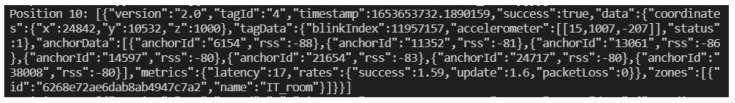
JSON Message carrying tag data.

**Figure 5 sensors-22-04946-f005:**
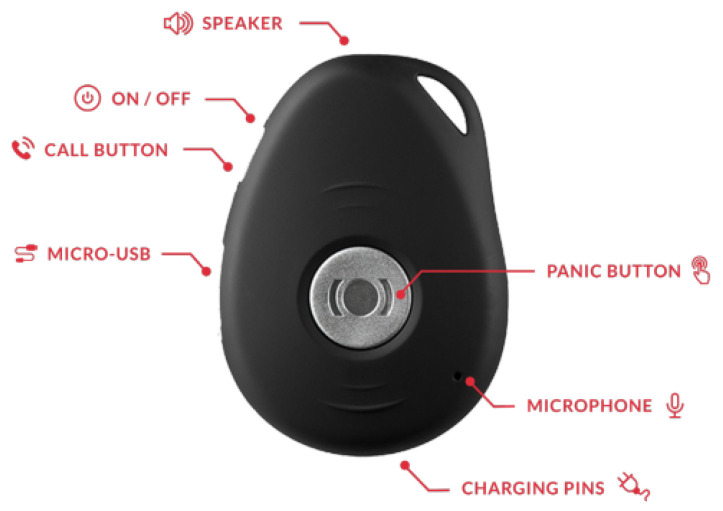
Hardware overview.

**Figure 6 sensors-22-04946-f006:**
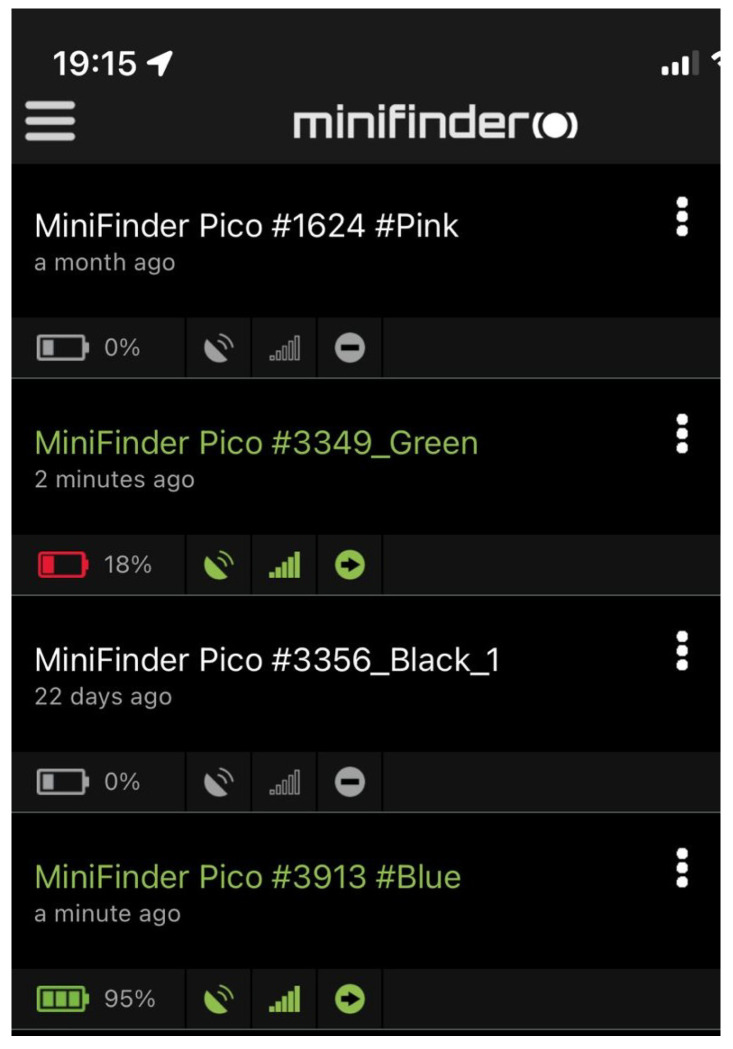
Device GPS unit status.

**Figure 7 sensors-22-04946-f007:**
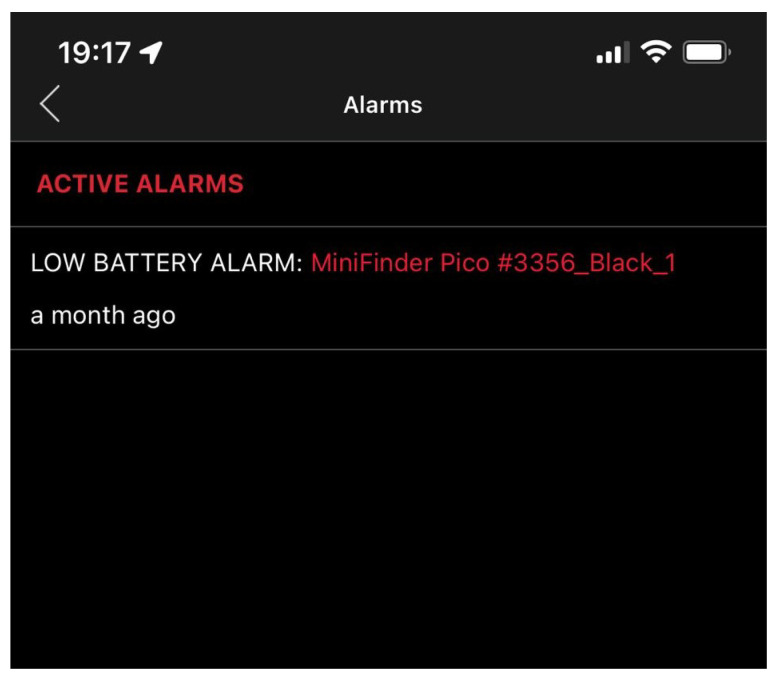
Low battery alarm.

**Figure 9 sensors-22-04946-f009:**
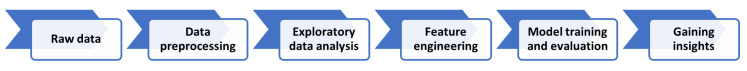
Machine learning process.

**Figure 10 sensors-22-04946-f010:**
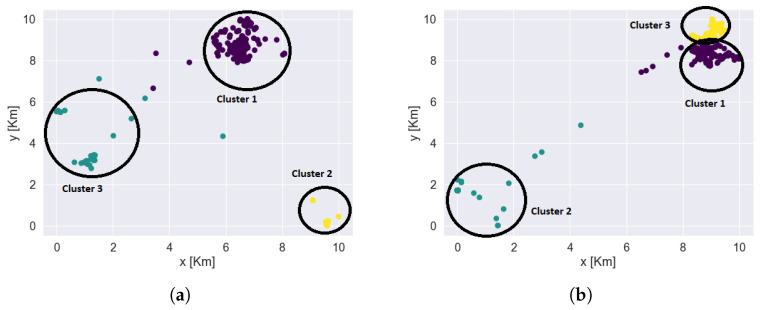
Clusters identified for Users. (**a**) User1; (**b**) User4.

**Figure 11 sensors-22-04946-f011:**
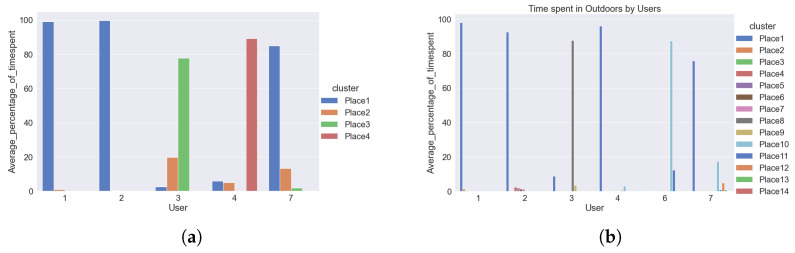
Percentage of time spent by users indoors and outdoors. (**a**) Indoor; (**b**) Outdoor.

**Figure 12 sensors-22-04946-f012:**
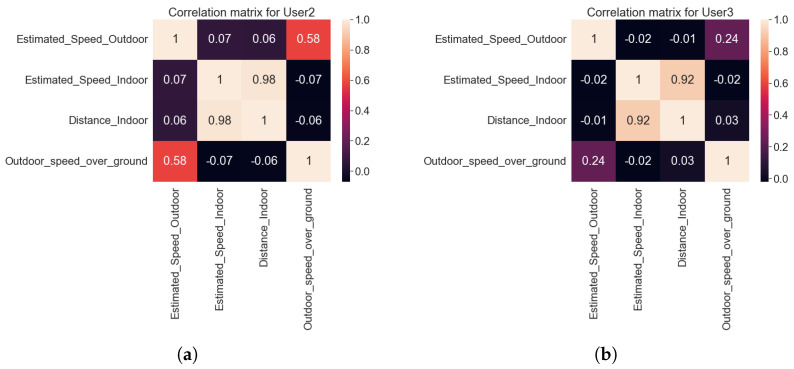
Correlation matrix. (**a**) User 2; (**b**) User 3.

**Figure 13 sensors-22-04946-f013:**
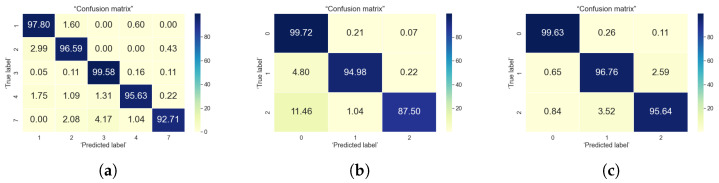
Confusion matrix for XGBoost algorithm using indoor data. (**a**) User classification; (**b**) UCLA based classification; (**c**) Lubben-based classification.

**Figure 14 sensors-22-04946-f014:**
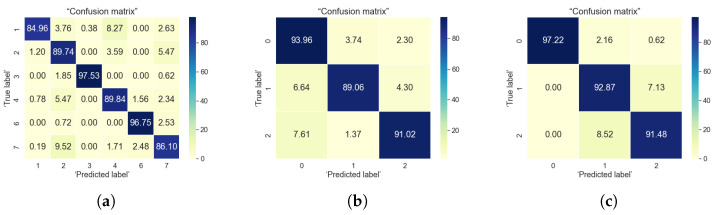
Confusion matrix for XGBoost algorithm using outdoor data. (**a**) User classification; (**b**) UCLA based classification; (**c**) Lubben-based classification.

**Figure 15 sensors-22-04946-f015:**
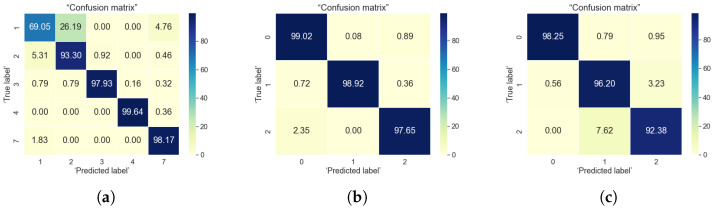
Confusion matrix for XGBoost algorithm using combined indoor and outdoor data. (**a**) User classification; (**b**) UCLA based classification; (**c**) Lubben-based classification.

**Table 1 sensors-22-04946-t001:** Previous studies comparison with current study.

Reference	Data	Sensors	Methodology	Findings	Results
[1]	ELSA dataset (The English Longitudinal Study of Ageing)	-	Machine learning analysis (XGBoost, LightGBM)	Predicting loneliness	AUC (Area under curve) 0.84–0.88
[2]	Health parameters	Fitbit watch	mobility patterns, Machine learning analysis	IOT platform for elderly monitoring	-
[6]	Health parameters	Accelerometer, ECG	Machine learning analysis (Naive Bayes)	Human activity recognition	Accuracy 0.92 (Fall detection), 0.99 (Resting), 0.99 (Walking)
[7]	Questionnaire on demographics, mobility patterns, use of public spaces, neighbourhood	-	Path analysis	Identifying loneliness using public space use and mobility patterns	-
[8]	Demographic, physical activity, health parameters	-	ANOVA, chi-square test	Physical activity intervention in loneliness	-
Current study	Sensor based mobility patterns	Pozyx, Pico minifinder	Machine learning analysis (XGBoost, Random forest, Support vector machine)	Identifying risk of loneliness using mobility patterns	Accuracy 0.90–0.98

**Table 2 sensors-22-04946-t002:** MQTT message structure.

Name	Data Type	Description
tagID	String	Tag which is captured
timestamp	Number	time of data capture (epoch time)
success	Boolean	indicating whether the position could be measured or not
data.coordinates	Array	The 3D location of the tag within the monitored region (local coordinates)
data.tagData.blinkIndex	Number	Index identifying the signal that the tag sent out.
data.tagData.accelerometer	Array	An array of acceleration measurements along *X*, *Y* and *Z* axes.
data.tagData.status	Number	The programmed state of the tag changeable with the interactive push button.
data.anchorData	Array	An array of the anchors which participated in detecting the tag location, and the RSS of the UWB packet at each anchor’s antenna.
data.zones	Array	Array containing the zone id and the zone name where the tag is present.

**Table 3 sensors-22-04946-t003:** Colour indicators and Device status.

Colour Indicators	Device Status
Green	Online state (location data is being transmitted)
Orange	Passive state (location data was sent over 10 min ago)
Red	Passive state (location data was sent over 1 h ago)
White	Offline state

**Table 4 sensors-22-04946-t004:** NMEA message structure.

Name	Data Type	Description
200637	Number	Time Stamp
A	String	Validity—A-OK, V-invalid.
6128.2257	Number	Current Latitude
N	String	North/South
02346.8965	Number	Current Longitude
E	String	East/West
0.54	Number	Speed in knots
126	Number	True course
280522	Number	Date Stamp

**Table 5 sensors-22-04946-t005:** Classification labels.

Name	Loneliness Labels
Low level of loneliness	0
Medium level of loneliness	1
High level of loneliness	2

**Table 6 sensors-22-04946-t006:** Result of machine learning algorithms on indoor and outdoor data.

	Indoor Data Results			Outdoor Data Results			Combined Data Results		
Algorithm	User Classification	UCLA Based Classification	Lubben Based Classification	User Classification	UCLA Based Classification	Lubben Based Classification	User Classification	UCLA Based Classification	Lubben Based Classification
Support vector machines	89.1%	85.1%	93.8%	77.7%	73.8%	84.6%	81.3%	97.0%	87.2%
Random forest	89.0%	82.3%	93.4%	89.4%	90.6%	93.4%	90.8%	98.2%	94.4%
XGBoost	96.4%	94.0%	97.3%	90.8%	91.3%	93.8%	91.6%	98.5%	95.6%

## Data Availability

Not applicable.

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
