# Peer review of "A Machine-Learning-Based Analysis of the Relationships between Loneliness Metrics and Mobility Patterns for Elderly"

_sensors, 2022, doi:10.3390/s22134946_

Round 1
Reviewer 1 Report
The authors present the article entitled “Machine-learning-based analysis of the relationships between loneliness metrics and mobility patterns for elderly”.
The main goal of this paper has been to find correlations (or features) relating the sensor-harnessed data with quantitative indicators or metrics which play an important role in identifying loneliness or social isolation.
The article presents the following concerns:
- Avoid using first-person sentences.
- Use an instead third person or passive voice sentences.
- The objective of the article is not all impressive. I suggest the authors to restructure the objective of the article by highlighting the novelty and contribution of the article (lines 63-64)
- Please define UCLA
- Figure 1: Use labels in the figure to identify each element of the system.
- Figure 3: Please pinpoint the location of the system.
- Lines 254-267: I suggest presenting this information as a table. Same with 320-328.
- Table 3: Are these results accurate? How were the results calculated? It is necessary to explain the method used to calculate those values.
- Include a table that compares the findings of the work vs the already reported in the state of the art.
Line 333-337 could be justified with the following machine learning algorithms references: A study of computing zero crossing methods
and an improved proposal for emg signals; A novel method for measuring subtle alterations in pupil size in children with congenital strabismus; Teaching challenges in covid-19 scenery: teams platform-based student satisfaction approach; Public space accessibility and machine learning tools for street vending spatial categorization
The following misspelling should be checked:
- line 3: “presence of disabilities…” should be rewritten as “The presence of disabilities…” Be careful with the articles in all text.
- line 10: “travelled…” should be rewritten as “traveled…”
- line13: “algorithm achieve…” should be rewritten as “algorithm achieved…”
- line 40: “which have proposed…” should be rewritten as “that have proposed…”
- line 271: “light weight…” should be rewritten as “ lightweight…”
- line 362: “mention…” should be rewritten as “mentioned…”
- line 367: “as apart…” should be rewritten as “as a part…”
- line 378: “overview on the average…” should be rewritten as “an overview of the average…”
- lines 385-386: “We used k-means clustering algorithms to identify the areas visited frequently by each user” should be rewritten as “We used k-means clustering algorithms to identify each user's frequently visited areas”
- line 404: “one retrieved…” should be rewritten as “one was retrieved…”
Reviewer 2 Report
[Comment 1] Novelty and literature review
[Subcomment 1a] (Section 1) Please clearly state how this study differs from [3].
[Subcomment 1b] The authors must list previous studies on loneliness factor analysis, then compare these studies with the authors' current study in a table.
[Subcomment 1c] It is necessary to address related previous studies about feeling monitoring to show what devices and technologies were used to measure the participants' feelings. Please add such explanations in the literature review. It is important to assess whether the button pushing device is the best method or not.
[Subcomment 1d] (lines 582-583) The authors need to mention the reason why they chose xgboost, random forest, and support vector machine. Please provide supporting references.
[Comment 2] Data and numerical experiments
[Subcomment 2a] I suggest the authors upload their data on an online repository and share the link in the manuscript, to allow reproducibility by other researchers.
[Subcomment 2b] I question whether the amount of collected data is sufficient to conduct the machine learning run. Please elaborate.
[Subcomment 2c] The authors need to report how well the participants use the devices (whether appropriate data were collected continuously during the measurement time or not).
[Subcomment 2d] (lines 402-404) Please mention the user IDs that had their data removed. I do not understand why the authors keep reporting such results if the data are removed.
[Subcomment 2e] (Section 4.4) I think the authors need to restate the input and output data used for the machine learning method here for clarity.
[Subcomment 2f] (Section 5) The results are not well presented. The authors must analyze the values of the input data and contrast them with the classification results to provide better intuition on the results for the readers. A question to answer is like: "People who are lonely have been proven to have more ... or less ...". If necessary, please present the analysis in a table.
[Comment 3] Writing quality and clarity
[Subcomment 3a] (Last line in Abstract) How different is "both indoor and outdoor data" from "combined indoor-outdoor data"? Please state it clearly.
[Subcomment 3b] (Section 3) Please add any reference about the project if exists.
[Subcomment 3c] All figures and tables must be mentioned in the text. I could not see Figure 14 mentioned anywhere yet.
[Subcomment 3d] Please differentiate the figure captions for Figures 13-15.
Round 2
Reviewer 2 Report
Thank you for your revisions.